# Does School Racial Composition Explain Why High Income Black Youth Perceive More Discrimination? A Gender Analysis

**DOI:** 10.3390/brainsci8080140

**Published:** 2018-07-30

**Authors:** Shervin Assari

**Affiliations:** 1Department of Psychiatry, University of Michigan, Ann Arbor, MI 48104, USA; assari@umich.edu; Tel.: +1-734-232-0445; Fax: +1-734-615-8739; 2Center for Research on Ethnicity, Culture and Health, School of Public Health, University of Michigan, Ann Arbor, MI 48104, USA; 3Department of Psychology, University of California, Los Angeles (UCLA), Los Angeles, CA 90095, USA; 4BRITE Center for Science, Research and Policy, University of California, Los Angeles (UCLA), Los Angeles, CA 90095, USA

**Keywords:** African Americans, socioeconomic status (SES), income, mental health, inter-group contact, discrimination

## Abstract

Recent research has documented poor mental health among high socioeconomic status (SES) Blacks, particularly African American males. The literature has also shown a positive link between SES and perceived discrimination, suggesting that perceived discrimination may explain why high SES Black males report poor mental health. To better understand the role of contextual factors in explaining this pattern, we aimed to test whether school racial composition explains why high income Black youth perceive more discrimination. We explored these associations by ethnicity and gender. Using data from the National Survey of American Life-Adolescent supplement (NSAL-A), the current study included 810 African American and 360 Caribbean Black youth, with a mean age of 15. Ethnicity, age, gender, income-to-needs ratio (SES), skin color, school racial composition, and perceived (daily) discrimination were measured. Using Stata 15.0 (Stata Corp., College Station, TX, USA), we fitted seven structural equation models (SEMs) for data analysis in the pooled sample based on the intersection of ethnicity and gender. Considerable gender by ethnicity variations were found in the associations between SES, school racial composition, and perceived discrimination. For African American males but not African American females or Caribbean Black males or females, school racial composition fully mediated the effect of SES on perceived discrimination. The role of inter-racial contact as a mechanism for high discrimination and poor mental health of Black American adolescents may depend on their intersection of ethnicity and gender. School racial composition may be a mechanism for increased perceived discrimination among high SES African American males.

## 1. Introduction

Although traditional research has mostly focused on the protective effects of high socioeconomic status (SES) on population health [1,2,3], recent research has documented poor mental health among Blacks of high SES [4,5]. While state-of-the-art studies have shown that SES indicators such as income protect populations against poor mental health [6,7,8,9,10,11], and the SES -health link is reported as “enduring, consistent, and growing” [12], there is some growing research that shows education [13], income [14,15], marital status [16,17,18,19], and employment [20] have diminished effects on the health of Blacks compared to Whites [4,5]. Although high SES is expected to increase human and material resources, reduce exposure to risk, and expand buffers that can mitigate the consequences when they occur [21,22,23,24], high SES is shown to increase perceived discrimination for Blacks [25,26]. This is particularly important because experiences of discrimination diminishes the health gains that follow SES resources among Blacks [27].

While high SES is protective against poor health overall [21,22,23,24], the health gain associated with high SES is diminished for Blacks [28] and Latinos [29]. The effects of education and income on a wide range of health behaviors such as drinking [13], diet [30], impulse control [18], body mass index [17], poor sleep [31], oral health [28,29], and chronic disease [14,32] are shown to be smaller for Blacks than Whites. Education attainment [13] and employment [20] have smaller effects on life expectancy for Blacks than for Whites. These patterns are robust as they are shown for youth [17,18,19,32], adults [17,20,30], and older adults [15], and have been replicated in cross-sectional [32] and longitudinal [17,20] studies.

Not only do Blacks gain less health from their SES than Whites [4,5], multiple recent studies have documented poor mental health of individuals with high SES Blacks [33,34,35]. That is, upward social mobility may be associated with additional psychological costs for Blacks [13,20,31,33,34,36,37]. Such a pattern will result in a very high gap in health of high SES Blacks and Whites. One study showed health disparities in 10 of the 16 health-related outcomes between Blacks and Whites with income of $175,000 and above [38]. In a 25-year longitudinal study of a national representative sample, Black men with high education credentials reported an increase in depressive symptoms over the course of follow up; a phenomenon which could not be observed for White males, White females, or Black women [33]. Among African American males, high household income is a risk factor for lifetime, 12-month, and 30-day major depressive disorder (MDD), a finding that was absent for African American females, and Caribbean Black males and females [39]. In another study, high income was associated with higher risk of MDD in African American men but not women [40]. In another study using a nationally representative sample, high education was positively associated with a risk of suicidal ideation among Caribbean Black females but not Caribbean Black males or African American males or females [34]. So, the role of high SES as a risk factor of depression for adults [40] and youth [39] is established by multiple studies [41].

Discrimination (actual and perceived) is reported as a potential explanation for the diminished health gain of Blacks from their SES resources [4,27]. However, whether or not high SES increases perception of discrimination is not consistent across all sub-groups of Blacks [42]. A recent study revealed considerable gender by ethnicity variations in the patterns of the associations between SES indicators and perceived discrimination. High family income was associated with high perceived discrimination in African American males and Caribbean Black females, however, SES indicators were not related to perceived discrimination for African American females or Caribbean Black males. Authors concluded that whether SES reduces or increases perceived discrimination among Black youth depends on the intersection of ethnicity and gender [42].

Most previous research has ignored within-Black heterogeneities due to ethnicity. Caribbean Black youth are among the most understudied subgroups of Black youth in the U.S. Caribbean Black and African American families differ in their history, life experiences, culture, norms, values, as well as socioeconomic status [43]. For instance, Caribbean Black Americans are mostly recent immigrants with higher employment, education, and income, compared to African Americans. They also differ regarding the history of slavery. Caribbean Blacks reside in Florida, New York, and New Jersey, which is different from African Americans who are spread across the United States, with greater concentration in the South [44,45].

Based on inter-group contact theory [46], inter-group contact is one of the mechanisms by which minority groups are exposed to prejudice and perceived discrimination. Several studies have suggested that Blacks who live in proximity to Whites may report more discrimination [25,26]. Negative intergroup contact ranges from mildly unpleasant interactions (e.g., awkwardness between strangers) to more severe incidents (e.g., verbal or physical abuse). Although less frequent than positive contact [47,48,49], negative contact with advantaged group members is relatively common for disadvantaged group members [50]. Moreover, one form of negative contact for minority group members is exposure to prejudice and discrimination [51,52]. In longitudinal studies of Black youth over 18 years, high SES youth and those who lived in predominantly White areas reported more discrimination and depression, and the effect on depression was fully mediated by discrimination [25,26]. This increased contact with Whites has been proposed to explain high perceived discrimination of high SES Blacks [25].

To better understand the role of contextual factors [25,26] in explaining why high SES Blacks are vulnerable to discrimination [25,26] and depression [39], this study used an ethnically diverse national sample of Black youth to investigate the associations between SES, school racial composition, and perceived discrimination by ethnicity and gender. We specifically hypothesized that school racial composition would explain why high income male Black youth perceive more discrimination.

## 2. Methods

### 2.1. Design

This cross-sectional study used data from the National Survey of American Life-Adolescents (NSAL-A) supplement [53,54,55]. NSAL was conducted as a part of the Collaborative Psychiatric Epidemiology Surveys (CPES), supported by the National Institute of Mental Health (NIMH).

### 2.2. Ethical Considerations

The NSAL (including NSAL-A) study protocol was approved by the Institute Review Board (IRB) at the University of Michigan (UM), Ann Arbor, Michigan. All participating adolescents provided assent. All adolescents’ legal guardians provided informed consent. Respondents received a financial compensation of $50 for their time.

### 2.3. Participants and Sampling

This study included 810 African American and 360 Caribbean Black youth aged from 13 to 17, with a mean age of 15 years old (standard deviation (SD) = 1.42). The NSAL-A sample was drawn from the NSAL household national probability sample of adult Blacks in the United States. In the first step, the NSAL-Adult households were screened for eligible Black (African American or Caribbean Black) adolescents living in the same households. Adolescents living in the same households were then randomly selected for participation. When more than one eligible adolescent was available in the household, two adolescents were selected based on the gender of the first eligible adolescent. Due to this sampling decision, the NSAL-A sample is non-independent. To address this issue, the adolescent supplement data were weighted to adjust for non-independence of the selection probabilities as well as household and individual levels. At the final step, the weighted data were post-stratified so the proportions would represent national estimates by age, gender, and ethnicity [56,57].

### 2.4. Interviews and Data Collection

All the NSAL interviews were conducted in English (82% face-to-face and 18% by phone). Computer-assisted personal interviews (CAPI) were applied for all the face-to-face interviews. In CAPI, respondents use a computer to answer the questions. CAPI is believed to improve data quality when questionnaires are long and complex [58]. Interviews lasted 100 min on average. The NSAL-A response rate was above 80%.

### 2.5. Measures

The study measured ethnicity, age, gender, SES (income to needs ratio), racial composition of school, and perceived daily discrimination.

*Ethnicity.* Ethnicity was self-identified and based on the ethnicity of the household in which the adolescent lived. Participants self-identified either as African Americans or as Caribbean Blacks. African American ethnicity was defined as being Black without any ancestral ties to the Caribbean. Caribbean Black ethnicity was defined as being Black and having ancestral ties to the following countries: Antigua and Barbuda, Barbados, Bahamas, Cuba, Dominican Republic, Dominica, Grenada, Haiti, Jamaica, Saint Vincent and the Grenadines, Trinidad and Tobago, Saint Lucia, and Saint Kitts and Nevis.

*Family Socioeconomic Status (SES).* Family SES was measured using income-to-needs ratio [59]. To measure the income-to-needs ratio, participants’ parents/guardians were asked about their family income using self-reported data. The income-to-needs ratio was calculated by dividing family income to number of individuals in the household. A higher income-to-needs ratio reflected higher SES. Income-to-needs ratio was measured in 6 levels [39,42,60].

*Skin Tone.* Self-reported skin tone (skin complexion) was measured using a single item measure. Participants reported their skin tone as one of the following five categories: 0 (“very light brown”), 1 (“light brown”), 2 (“medium”), 3 (“dark brown”), and 4 (“very dark brown”). This measure strongly correlates with interviewers’ rating of skin color (correlation coefficient = 0.80), indicating high validity of self-report measure. Higher score was reflective of darker skins [61,62].

*Racial Composition of School.* The racial composition [63,64,65,66] of schools was measured using the following items: “Think about the other students in most of your classes. Would you say that almost all, very many, some, a few, or (none/no other) (are/were): Black/African American students, Latino/Hispanic students, Asian students, and White students.” Different questions were asked for Black/African American, Latinos /Hispanic students, Asian students, and White students. Responses were (1) None, (2) A Few, (3) Some, (4) Very Many, and (5) Almost All.

*Perceived Discrimination.* Perceived (daily) discrimination was measured using a 13-item measure. This was a modified version of the Everyday Discrimination Scale (EDS), developed by David Williams [67]. The items assess chronic, routine, and less overt discrimination rather than acute, major, overt experiences. The measure items asked the individuals whether any of the discriminatory events have occurred over the past year. Sample items include: “being followed around in stores”, “people acting as if they think you are dishonest”, “receiving poorer service than other people at restaurants” and “being called names or insulted”. Although the original measure only included 10 items, three additional items were added to reflect discrimination at school (teacher discrimination). Although the EDS measure was originally developed and normed among adults, it has been widely used for youth [61,67,68,69]. The responses were on a Likert scale ranging from 1 (never) to 6 (almost everyday). A sum score was calculated, which reflected frequency of discriminatory exposures over the past year (α = 0.86).

### 2.6. Statistical Analysis

To accommodate the NASL-A complex sampling design, Stata 15.0 (Stata Corp., College Station, TX, USA) was used for data analysis. This software allowed us to recalculate the complex design-based variance and standard errors. As a result, all proportions and inferences reflect the NSAL-A’s complex design and are nationally representative. Seven structural equation models (SEMs) were used for multivariable analysis [70,71,72]. Income-to-needs ratio (SES) was the independent variable. The dependent variable was perceived (daily) discrimination. Age and skin color were the covariates. We controlled for skin tone because it may increase perceived discrimination for Black youth [61]. School racial composition was the mediator. Direct paths were assumed from independent variables and covariates to the mediator the percent of Whites in schools) and outcome (perceived discrimination). We also allowed a path from mediator to outcome. Ethnicity and gender were the focal moderators. We used full information maximum likelihood to handle the missing data.

*Model 1* was tested in the pooled sample, with ethnicity, age, gender, and skin color as covariates. *Model 2* and *Model 3* were estimated in each ethnic group, with gender, age, and skin color as the covariates. *Model 4* to *Model 7* were conducted in each ethnicity by gender group. 

Models with and without constrained paths across groups were estimated. For our final model, we released constraints as we did not gain improvement in goodness of fit in models with constrains. Fit statistics included chi-square, the comparative fit index (CFI) (>0.90), the root mean squared error of approximation (RMSEA) (<0.06), and chi-square to degrees of freedom ratio [73,74,75]. Unstandardized path coefficients, standard error, 95% confidence interval (CI), z, and p values were reported.

## 3. Results

Table 1 describes age, SES (family income, and income-to-need ratio (poverty index)), and perceived discrimination in the pooled sample, as well as across ethnic by gender groups. The highest level of financial hardship was reported by Caribbean Black females. The highest level of discrimination was reported by Caribbean Black males.

Table 2 summarizes the results of seven SEMs with perceived discrimination as the outcome, family income to needs ratio as the independent variable, age as the covariate, and school racial composition as the mediator. *Model 1* was estimated in the pooled sample. *Model 2* and *Model 3* were estimated in African Americans and Caribbean Blacks. *Model 4* to *Model 7* were conducted in each ethnicity by gender groups (Table 2, Figure 1 and Figure 2).

In the pooled sample, higher family income-to-needs ratio was associated with higher percentage of Whites at school which was in turn associated with more perceived discrimination. In the pooled sample, however, percentage of Whites at school did not explain the effect of family income on perceived discrimination. Ethnic by gender differences were found in the associations between family income, percentage of Whites at school, and perceived discrimination. For African American males but not African American females or Caribbean Black males or females, school racial composition fully mediated the effect of SES on perceived discrimination (Table 3, Figure 1 and Figure 2).

## 4. Discussion

Borrowing data with a national sample, the current study explored ethnic by gender heterogeneity in the pattern of associations between SES, school racial composition, and perceived discrimination among Black youth. School racial composition mediated the effect of SES on perceived discrimination for male African American but not female African American and males or female Caribbean Black youth.

Although this study is not the first to investigate the positive association between SES and perceived discrimination among Blacks [25,33], it is one of the first studies to suggest that the percentages of Whites in school may play a role in explaining why high SES African American boys report more discrimination. In a recent study, Black youth who had higher SES had a higher likelihood of living in predominantly White areas and experienced more discrimination and depression. Interestingly, in those studies, discrimination fully explained why depression was more common for such youth [25,26].

Considerable research suggests that high SES increases exposure to discrimination for Blacks [25,26]. There is even some research showing that high SES may also increase sensitivity (vulnerability) to discrimination, measured as a stronger association between discrimination and MDD in those with high subjective SES [25,27]. While high SES increases the effect of discrimination on depression [25], discrimination may reduce the health gain that follows high SES [27]. A study, however, failed to show discrimination as a mediator of effect of SES on depression for Black men [76].

How discrimination contributes to the poor mental health of Black youth is complex and may be a function of the intersection of ethnicity, gender, and SES. Similarly, the underlying mechanisms for health disparities among Blacks are complex and at least some are due to low health gain from SES and some from risk associated with high SES [5].

### 4.1. Direction for Future Research

There is still a need to study the role of perceived discrimination as a potential explanatory mechanism for poor mental health of high SES Blacks, particularly males. There is also a need to investigate moderators and mediators of such effects. Culture, values, social norms, socialization, attribution style, vigilance, personality, racial and ethnic identity, and coping may all have some role in explaining the extra vulnerability of high SES Black youth to discrimination. Future research could test whether family type, family processes, social support, race socialization, explain these effects. More research is also needed on other contextual factors such as poverty, density of Blacks, and frequency of contacts across groups.

Future research is needed on contextual and individual level factors that shape exposure and vulnerability to perceived discrimination. Future research should also examine what proportion of these findings are due to actual discrimination and what percentage is due to vigilance and attribution of ambiguous exposures to race.

### 4.2. Theoretical Implications

These findings may have implications for expanding the existing theoretical knowledge on the role of race and SES in shaping health disparities. While traditional frameworks such as Fundamental Cause and Social Determinants of Health theories [21,22,23,24] focus on the health gain rather than the psychological costs of upward social mobility, health disparities researchers should be aware of instances that high SES becomes a risk factor. The effects of high SES, however, are not limited to protective factors. Whether SES operates as a risk or as a protective factor depends on demographic factors (sub-population), place, context, social structure, SES indicator, and the outcome. At least in the US, there are some hidden costs to high SES for Black youth. More work is needed on how high SES operates as an extra risk factor and the role of perceived discrimination for Black youth.

### 4.3. Limitations

To interpret our findings, a full consideration should be given to our study limitations. First, our study had a cross-sectional design, causative inferences depend on how well the models represent the causal processes under investigation [77]. Second, the study only controlled for age and skin color as the confounders, and important variables such as racial identity, vigilance and race socialization were not included. Third, the study only included income as the SES indicator. Other SES factors such as parental education, family structure, living place, and also other contextual factors such as the density of Whites in the neighborhood were not investigated. Despite these limitations, these findings make a unique contribution to the literature on SES, gender, and discrimination in Black youth, as it helps us understand why high SES may operate as a vulnerability factor among Blacks. Before further interpretation, there is a need to replicate these findings using other datasets, settings, cohorts, and age groups [78,79,80,81].

## 5. Conclusions

For African American male youth, high SES may be associated with higher perceived discrimination because of attending schools with more Whites. This finding help us understand why high SES may operate as a vulnerability factor for Black males, and why high SES increases exposure to discrimination and risk of depression. 

## Figures and Tables

**Figure 1 brainsci-08-00140-f001:**
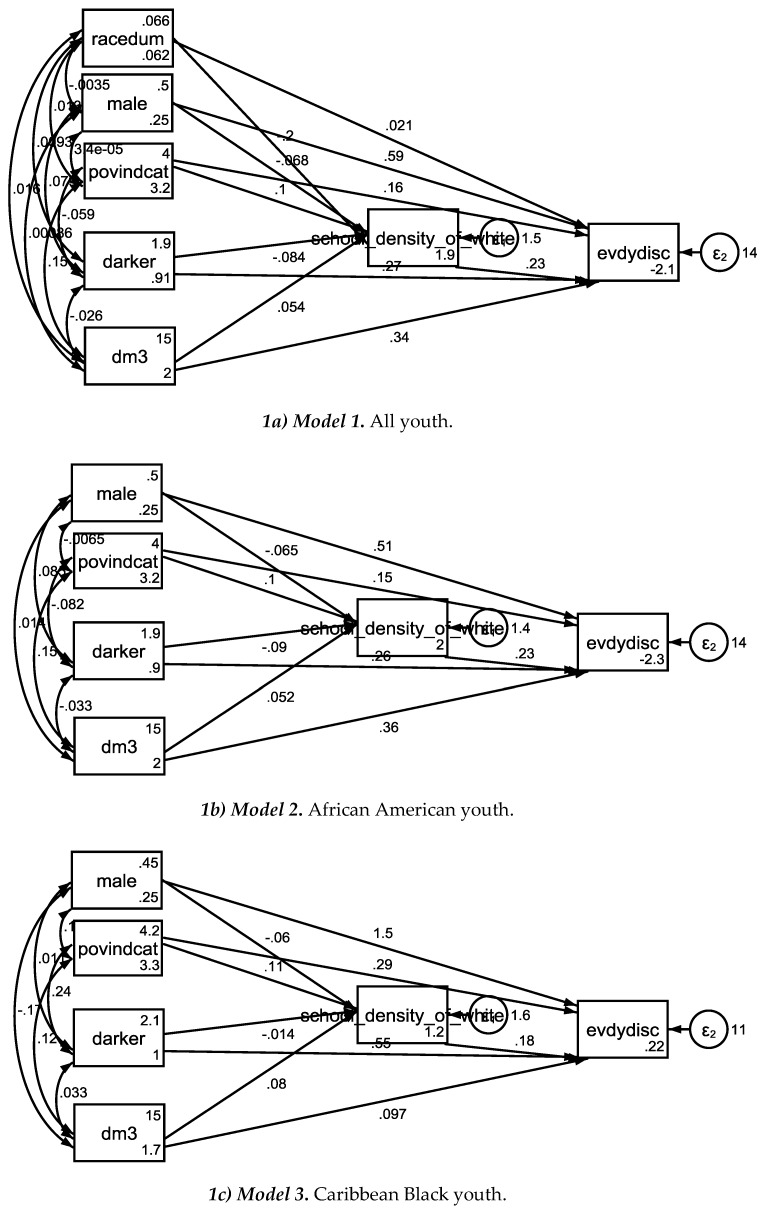
Model 1 to Model 3 in the pooled sample, African American, and Caribbean Black youth.

**Figure 2 brainsci-08-00140-f002:**
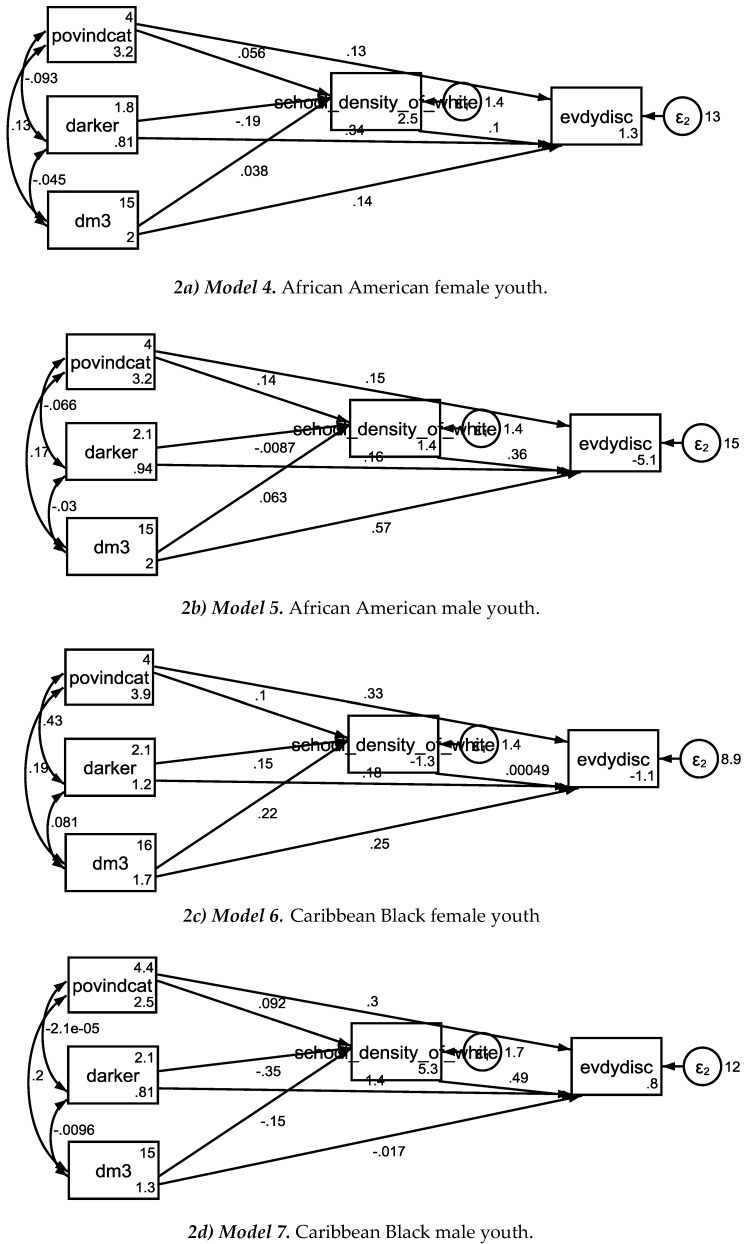
Model 4 to Model 7 in the pooled sample, African American, and Caribbean Black youth.

**Table 1 brainsci-08-00140-t001:** Descriptive statistics.

	All		African American Female		African American Male		Caribbean Black Female		Caribbean Black Male	
	Mean	95% CI	Mean	95% CI	Mean	95% CI	Mean	95% CI	Mean	95% CI
Age (Years)	14.97	14.84–15.09	14.91	14.72–15.10	14.99	14.83–15.15	15.55	15.44–15.66	14.80	14.59–15.01
Family Income (Centered)	170.31	−4159.66–4500.27	196.85	−4874.99–5268.70	83.65	−6101.89–6269.19	−478.97	−8941.67–7983.74	1930.03	−7151.01–11,011.08
Income to Needs Ratio	3.98	3.73–4.23	3.98	3.74–4.21	3.95	3.58–4.33	3.99	3.614.38	4.43	3.58–5.27
Perceived Discrimination (Everyday)	5.07	4.68–5.47	4.76	4.31–5.21	5.36	4.81–5.91	4.48	3.75–5.22	6.13	4.25–8.01

Confidence interval (CI).

**Table 2 brainsci-08-00140-t002:** Path coefficients for Model 1 to Model 3 in the pooled sample, African American, and Caribbean Black youth.

Independent Variable	Dependent Variable	*b*	(SE)	95% CI		*z*	*p*
***Model 1 (All)***							
Independent Variable:							
Income-to-needs ratio	School Density	0.10	0.02	0.05	0.15	4.10	0.000
Skin Color (Dark)	School Density	−0.08	0.05	−0.18	0.01	−1.74	0.082
Age	School Density	0.05	0.03	−0.01	0.11	1.73	0.083
Ethnicity (CB)	School Density	−0.20	0.16	−0.50	0.11	−1.27	0.203
Gender (Male)	School Density	−0.07	0.09	−0.25	0.11	−0.75	0.454
Intercept	School Density	1.94	0.49	0.99	2.90	3.99	0.000
School density (Percentages of Whites)	Perceived Discrimination	0.23	0.12	0.00	0.46	1.96	0.050
Income-to-needs ratio	Perceived Discrimination	0.16	0.08	0.01	0.31	2.11	0.034
Skin Color (Dark)	Perceived Discrimination	0.27	0.15	−0.01	0.56	1.88	0.060
Age	Perceived Discrimination	0.34	0.10	0.15	0.53	3.58	0.000
Ethnicity (CB)	Perceived Discrimination	0.02	0.33	−0.62	0.66	0.06	0.950
Gender (Male)	Perceived Discrimination	0.59	0.28	0.05	1.13	2.13	0.033
Intercept	Perceived Discrimination	−2.11	1.47	−4.99	0.77	−1.44	0.151
***Model 2 (African Americans)***							
Income-to-needs ratio	School Density	0.10	0.03	0.05	0.15	3.85	0.000
Skin Color (Dark)	School Density	−0.09	0.05	−0.19	0.01	−1.78	0.075
Age	School Density	0.05	0.03	−0.01	0.12	1.62	0.106
Gender (Male)	School Density	−0.07	0.09	−0.25	0.12	−0.69	0.487
Intercept	School Density	1.98	0.50	0.99	2.97	3.93	0.000
School density (Percentages of Whites)	Perceived Discrimination	0.23	0.12	−0.01	0.47	1.85	0.064
Income-to-needs ratio	Perceived Discrimination	0.15	0.08	−0.01	0.30	1.81	0.070
Skin Color (Dark)	Perceived Discrimination	0.26	0.16	−0.05	0.56	1.64	0.102
Age	Perceived Discrimination	0.36	0.10	0.17	0.56	3.62	0.000
Gender (Male)	Perceived Discrimination	0.51	0.29	−0.06	1.08	1.74	0.082
Intercept	Perceived Discrimination	−2.30	1.55	−5.33	0.73	−1.49	0.136
***Model 3 (Caribbean Blacks)***							
Income-to-needs ratio	School Density	0.11	0.08	−0.05	0.26	1.36	0.175
Skin Color (Dark)	School Density	−0.01	0.16	−0.32	0.29	−0.09	0.927
Age	School Density	0.08	0.10	−0.11	0.27	0.81	0.417
Gender (Male)	School Density	−0.06	0.30	−0.64	0.52	−0.20	0.840
Intercept	School Density	1.17	1.51	−1.80	4.13	0.77	0.440
School density (Percentages of Whites)	Perceived Discrimination	0.18	0.20	−0.21	0.56	0.90	0.367
Income-to-needs ratio	Perceived Discrimination	0.29	0.12	0.05	0.53	2.36	0.018
Skin Color (Dark)	Perceived Discrimination	0.55	0.24	0.08	1.01	2.31	0.021
Age	Perceived Discrimination	0.10	0.19	−0.28	0.47	0.51	0.613
Gender (Male)	Perceived Discrimination	1.54	0.51	0.55	2.54	3.04	0.002
Intercept	Perceived Discrimination	0.22	2.65	−4.97	5.42	0.08	0.933

**Table 3 brainsci-08-00140-t003:** Path coefficients for Model 4 to Model 7 in male and female African American and Caribbean Black youth.

Independent Variable	Dependent Variable	*b*	(SE)	95% CI		*z*	*p*
***Model 4 (African American Females)***							
Income-to-needs ratio	School Density	0.06	0.04	−0.01	0.13	1.58	0.114
Skin Color (Dark)	School Density	−0.19	0.07	−0.32	−0.05	−2.74	0.006
Age	School Density	0.04	0.05	−0.05	0.13	0.82	0.414
Intercept	School Density	2.53	0.75	1.07	4.00	3.40	0.001
School density (Percentages of Whites)	Perceived Discrimination	0.10	0.16	−0.22	0.42	0.62	0.537
Income-to-needs ratio	Perceived Discrimination	0.13	0.11	−0.08	0.34	1.20	0.230
Skin Color (Dark)	Perceived Discrimination	0.34	0.22	−0.08	0.77	1.59	0.112
Age	Perceived Discrimination	0.14	0.14	−0.13	0.41	0.99	0.321
Intercept	Perceived Discrimination	1.33	2.26	−3.09	5.75	0.59	0.555
***Model 5 (African American Males)***							
Income-to-needs ratio	School Density	0.14	0.04	0.07	0.22	3.86	0.000
Skin Color (Dark)	School Density	−0.01	0.07	−0.15	0.13	−0.12	0.902
Age	School Density	0.06	0.04	−0.02	0.15	1.45	0.147
Intercept	School Density	1.41	0.66	0.12	2.70	2.13	0.033
School density (Percentages of Whites)	Perceived Discrimination	0.36	0.19	−0.01	0.73	1.93	0.050
Income-to-needs ratio	Perceived Discrimination	0.15	0.12	−0.08	0.37	1.24	0.213
Skin Color (Dark)	Perceived Discrimination	0.16	0.22	−0.27	0.58	0.71	0.475
Age	Perceived Discrimination	0.57	0.14	0.30	0.85	4.05	0.000
Intercept	Perceived Discrimination	−5.14	2.09	−9.23	−1.05	−2.46	0.014
***Model 6 (Caribbean Black Females)***							
Income-to-needs ratio	School Density	0.10	0.09	−0.07	0.27	1.17	0.243
Skin Color (Dark)	School Density	0.15	0.16	−0.16	0.47	0.96	0.337
Age	School Density	0.22	0.12	−0.02	0.46	1.77	0.077
Intercept	School Density	−1.30	1.87	−4.97	2.36	−0.70	0.485
School density (Percentages of Whites)	Perceived Discrimination	0.00	0.22	−0.43	0.44	0.00	0.998
Income-to-needs ratio	Perceived Discrimination	0.33	0.13	0.07	0.60	2.51	0.012
Skin Color (Dark)	Perceived Discrimination	0.18	0.28	−0.38	0.73	0.62	0.535
Age	Perceived Discrimination	0.25	0.23	−0.19	0.70	1.12	0.261
Intercept	Perceived Discrimination	−1.13	3.29	−7.58	5.31	−0.34	0.730
***Model 7 (Caribbean Black Males)***							
Income-to-needs ratio	School Density	0.09	0.14	−0.18	0.36	0.67	0.502
Skin Color (Dark)	School Density	−0.35	0.24	−0.82	0.12	−1.45	0.148
Age	School Density	−0.15	0.12	−0.38	0.08	−1.25	0.210
Intercept	School Density	5.28	2.03	1.30	9.27	2.60	0.009
School density (Percentages of Whites)	Perceived Discrimination	0.49	0.35	−0.19	1.18	1.41	0.157
Income to needs ratio	Perceived Discrimination	0.30	0.26	−0.22	0.82	1.14	0.255
Skin Color (Dark)	Perceived Discrimination	1.37	0.39	0.62	2.13	3.57	0.000
Age	Perceived Discrimination	−0.02	0.30	−0.60	0.57	−0.06	0.954
Intercept	Perceived Discrimination	0.80	4.63	−8.26	9.87	0.17	0.862

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
