# Peer review of "Does School Racial Composition Explain Why High Income Black Youth Perceive More Discrimination? A Gender Analysis"

_brainsci, 2018, doi:10.3390/brainsci8080140_

Reviewer 1 Report

The current study has the potential to make a significant contribution to the research literature. However, in the present format there are significant changes that must me made.

Title:

The title is misleading in that I thought the original focus of the study was on African American males. The title needs to be inclusive of the other populations that are being examined in the study. I do understand the title being catchy...but it doesn't accurately reflect the populations being studied.

Abstract:

The abstract needs to be significantly revised. The mean age of the sample is missing. Please remove the sentences, "The dependent variable....to focal moderates". This belongs in the data analysis section. Again, I initially put why include Caribbean Black youth, but later found out that Caribbean Black adolescents (male and female) were included in the study. 

The last sentence is not clear. Please state what findings were found for the specific groups...using the term Black youth diminishes these findings. In addition, to be inclusive of the diversity within the Black adolescents...please use the term "Black American adolescents". 

Introduction:

There are several sentence fragments and grammatical errors throughout the manuscript. For instance on page 1, 2nd paragraph, "while high SES protective against..." - a word is missing. The same with the next sentence, "the effects of education and income..."

2nd page, "not only Blacks gain..." - there is a word missing. The sentence that starts of with, "such patterns...." needs to be revised as well. The sentence that starts off with, "among African American males...." - word missing.

Overall, the introduction is difficult to follow. Please review for typos and grammatical errors.

Additionally, the introduction did not address the unique differences between Caribbean Black and African adolescents boys and girls. Why study this? What are the unique differences between this group? What are the hypotheses based on previous literatures. 

There is also no guiding framework which also contributes to the lack of  clarity throughout the manuscript. 

Methods:

Please proofread the ethical considerations section. 

In the participants and sampling section the number of participants need to be reported in this section and the mean of adolescents age. What was the average income? 

Please revise and proofread the interviews and data collection section.

Measures

The SES section need to be explained in detail. As it is currently written, it is confusing how SES was measured and coded. Also, please provide citations on why this was coded this way. I would also review previous research studies using the NSAL-Adolescent data to see how other authors coded SES. 

The racial composition section is another section that is confusing. How was this coded? Why did the author choose to code it this way. Were other studies reviewed that used similar coding methods to assess school racial composition using the NSAL? 

Statistical Analyses Section

This section was also confusing. Similar to the sections, please proofread. If you were testing models then each step needs to be stated clearly. Model fit indices should be provided for each model. Additionally, the abstract states that age is the covariate, but model 1 states that ethnicity and gender are covariates. Please explain this further. How and why was school racial composition the mediator. What guiding framework does this stem from? What is the rationale? 

The biggest confusion in the model was the addition of skin color (dark) in the model. What does this mean? Where did this come from? How and why was this used? There was no mention of this before. 

As I result of the current analyses, I am not confident in the overall findings from this study. This paper needs significant work. 

Author Response

Thanks for the constructive comments.

Title:

Title is changes so it is not exclusively focusing on males. It is now more broad.

Abstract:

The abstract is significantly revised. The mean age of the sample is added to the abstract.

We removed the sentences, "The dependent variable....to focal moderates".

The last sentence of the abstract is revised. To be inclusive of the diversity within the Black adolescents we are now using the term "Black American adolescents". 

Introduction:

All the sentence fragments and grammatical errors are corrected throughout the manuscript. This includes added the missing words to:

"while high SES protective against..."

"the effects of education and income..."

"not only Blacks gain..."

"such patterns...." "among African American males...."

Overall, the introduction is now revised and has reduced typos and grammatical errors.

Now the introduction addresses the unique differences between Caribbean Black and African adolescents boys and girls. I mentioned the unique differences between this group. I added some information from the previous literatures. 

Guiding framework is inter-group contact. We added 6-7 references on this topic. And a paragraph.

Methods:

Please proofread the ethical considerations section. 

In the participants and sampling section, we added the number of participants need . We also added the mean adolescents age.

We checked the interviews and data collection section.

 Measures

The SES section is now better explained. We added a Census citation. We also mentioned several other NSAL papers that have used the same variable / coding for SES. 

We have provided citations for the paragraph “racial composition”. We could not find any previous studies using this variable in the NSAL.

 Statistical Analyses Section is also proofread. Age is added as the covariate.

We added one paragraph to our introduction that how and why was school racial composition the mediator. (Frequency of contact and exposure to Whites) . 

Sorry that skin color (dark) was not explained in the methods and statistics. Now it is added. We have added a paragraph that explains how it is measured. We also cited a NSAL study and other studies using the same measure / construct.

Reviewer 2 Report

Several pointed suggestions below:

Abstract: Recent research has documented poor mental health of high socioeconomic status SES Blacks
Abstract: Recent research has documented poor mental health among Blacks of high socioeconomic status (SES)
We explored this aim by ethnicity and gender.
We explored this aim by examining both ethnicity and gender.
Using data from the National Survey of American Life—Adolescent supplement (NSAL-A), current study
Using data from the National Survey of American Life—Adolescent supplement (NSAL-A), the current study
Role of inter-racial contact as a mechanism for high discrimination and poor mental health of Black youth depend on their intersection of ethnicity and gender,
Role of inter-racial contact as a mechanism for high discrimination and poor mental health of Black youth may depend on their intersection with ethnicity and gender,
p.1
there are some growing research that shows
 there is some growing research that shows
This is particularly important because experiences of discrimination on depression diminishes the effects of increased SES among Blacks [24].
This is particularly important because experiences of discrimination  diminishes the effects of increased SES on depression among Blacks [24]. [?]
While high SES protective against poor health overall [11–14]
While high SES is protective against poor health overall [11–14]
The effects of education and income on a wide range of health behaviors such as drinking [15], diet [74], impulse control [67], body mass index [66], poor sleep [16], oral health [72,73], and chronic disease [63,70].
The effects of education and income on a wide range of health behaviors such as drinking [15], diet [74], impulse control [67], body mass index [66], poor sleep [16], oral health [72,73], and chronic disease [63,70] have been documented.
p.2
Among African American males, high household income a risk factor for lifetime, 12-month, and 30-day major depressive disorder
Among African American males, high household income was found to be a risk factor for lifetime, 12-month, and 30-day major depressive disorder
p.1
So, role of high SES as a risk factor of depression is established by multiple studies [26] for adults [22] and youth [42].   
So, the role of high SES as a risk factor of depression is established by multiple studies [26] for adults [22] and youth [42].  
p.2
High family income (income to needs ratio) was associated with high (but not low) perceived discrimination
High family income (income to needs ratio) was associated with high perceived discrimination [Rev. conversely, low income is paired with low perceived discrimination, right?]
Authors concluded that whether SES reduces or increases perceived discrimination among Black youth depends on the intersection of ethnicity by gender
 Authors concluded that whether SES reduces or increases perceived discrimination among Black youth depends on the intersection of ethnicity with gender
before "We specifically hypothesized that school racial composition would explain why high income male Black youth perceive more discrimination." I see the need of one more line reasoning why school composition is expected to have such a role, from some prior study/theory.
NSAL was conducted a part of the Collaborative Psychiatric Epidemiology Surveys
NSAL was conducted as part of the Collaborative Psychiatric Epidemiology Surveys
All participating adolescents Assent provided assent.
All participating adolescents provided assent.
NSAL-A sample was drawn from NSAL household
The NSAL-A sample was drawn from the NSAL household
p.3
To measure income to needs ratio, participants parents/ guardians were asked about their family income was measured using self-reported data.
To measure income to needs ratio, participants parents/ guardians were asked about their family income using self-reported data.
Income to needs ratio was calculated by dividing family income to number of individuals in the household.
[Rev: this resembles more income per household member, so calling it household income or something in this vein would seem more appropriate; the needs part does not seem to have been easured to call it income to needs ratio]
p.4
Arrows were drawn from independent variable and covariates to the mediator (% Whites) and outcome (perceived discrimination).
Direct paths were assumed from independent variable and covariates to the mediator (% Whites) and outcome (perceived discrimination).
We used maximum likelihood to handle the missing data.
We used full information maximum likelihood to handle the missing data. [Rev. I assume it was FIML: the sign is if the N in analysis does not drop at all or not much]
Model 1 was ran in the pooled sample, with ethnicity and gender as covariates. Model 2 and Model 3 were estimated in each ethnic group, with gender as the covariate. Model 4 to Model 7 were conducted in each ethnicity by gender group.
Model 1 was tested on the pooled sample, with ethnicity and gender as covariates. Model 2 and Model 3 were estimated in each ethnic group, with gender as the covariate. Model 4 to Model 7 were conducted in each ethnicity by gender group.
Table 1
[was cut off on sides, cannot read edges]
In the pooled sample, higher family income to needs ratio was associated with higher % percentage of
In the pooled sample, higher family income to needs ratio was associated with higher percentage of
p.5

table 2
[I would mention that effects are to the 1st variable from the ones below it, or mark the outcoms as DVs with some superscripts]
p.7

Figures
[figures will definitely have to be re-drawn in Powerpoint, the easiest tool, some SEM such templates are online: the simplest, the better; I would BOLD the sig. paths, however, because one is tempted to compare across models the same paths, I suggest you enter the standardized numbers in the figures, and mark significance of the unstandardized estimates; make this clear in the figure notes]
p.9
Borrowing a national sample,
Borrowing information from a national sample,
it is one of the first studies that suggest % Whites at school
it is one of the first studies that suggest the percent of Whites in schools
Considerable research suggests that high SES increases exposure to discrimination for Blacks [].
Considerable research suggests that high SES increases exposure to discrimination for Blacks. [or add a reference]
While most traditional theories such as 31 Fundamental Cause Theory (FCT) and Social Determinants of Health (SDH) theories [11–14] focus 32 on the health gain rather than the psychological costs that follows high SES.
Most traditional theories such as 31 Fundamental Cause Theory (FCT) and Social Determinants of Health (SDH) theories [11–14] focus 32 on the health gain rather than the psychological costs that follows high SES
[the following sentence is hard to decipher/fix: "Conceptualization and theorization of The health disparities may benefit from instances that high SES becomes a risk factor."]
First, our study had a cross-sectional design, thus no causative inferences are allowed [52].

First, our study had a cross-sectional design, causative inferences depend on how well the models represent the causal processes under investigation [52].
[I would add some citations like
Pearl, J. (1998). Graphs, causality, and structural equation models. Sociological Methods & Research, 27(2), 226.
Pearl, J. (2011). The causal foundations of structural equation modeling. In R. H. Hoyle (Ed.), Handbook of structural equation modeling (pp. 68-91). New York, NY: The Guilford Press.]
p.10
[this line from Conclusions below should be rephrased and used twice before, in Abstract and at the begining of the Discussion, it's a very direct and intuitive statement of the findings, I was looking for one]
"for African American male youth, high SES may be associated with 52 higher perceived discrimination because of attending schools with more Whites"

Author Response

All of these are now corrected:

Before: Recent research has documented poor mental health of high socioeconomic status SES Blacks
After: Recent research has documented poor mental health among Blacks of high socioeconomic status (SES)

Before: We explored this aim by ethnicity and gender.
After: We explored this aim by examining both ethnicity and gender.
Before: Using data from the National Survey of American Life—Adolescent supplement (NSAL-A), current study 
After: Using data from the National Survey of American Life—Adolescent supplement (NSAL-A), the current study 

Before: Role of inter-racial contact as a mechanism for high discrimination and poor mental health of Black youth depend on their intersection of ethnicity and gender,
After: Role of inter-racial contact as a mechanism for high discrimination and poor mental health of Black youth may depend on their intersection with ethnicity and gender,
p.1 
Before: there are some growing research that shows 
After: there is some growing research that shows 
Before: This is particularly important because experiences of discrimination on depression diminishes the effects of increased SES among Blacks [24]. 
After: This is particularly important because experiences of discrimination diminishes the effects of increased SES on depression among Blacks [24]. [?]
Before: While high SES protective against poor health overall [11–14]
After: While high SES is protective against poor health overall [11–14]
Before: The effects of education and income on a wide range of health behaviors such as drinking [15], diet [74], impulse control [67], body mass index [66], poor sleep [16], oral health [72,73], and chronic disease [63,70].
 After: The effects of education and income on a wide range of health behaviors such as drinking [15], diet [74], impulse control [67], body mass index [66], poor sleep [16], oral health [72,73], and chronic disease [63,70] have been documented.
Before: Among African American males, high household income a risk factor for lifetime, 12-month, and 30-day major depressive disorder 
After: Among African American males, high household income was found to be a risk factor for lifetime, 12-month, and 30-day major depressive disorder 
Before: So, role of high SES as a risk factor of depression is established by multiple studies [26] for adults [22] and youth [42].   
After: So, the role of high SES as a risk factor of depression is established by multiple studies [26] for adults [22] and youth [42].  
Before: High family income (income to needs ratio) was associated with high (but not low) perceived discrimination
After: High family income (income to needs ratio) was associated with high perceived discrimination [Rev. conversely, low income is paired with low perceived discrimination, right?]
Before: Authors concluded that whether SES reduces or increases perceived discrimination among Black youth depends on the intersection of ethnicity by gender 
After: Authors concluded that whether SES reduces or increases perceived discrimination among Black youth depends on the intersection of ethnicity with gender 
Before: NSAL was conducted a part of the Collaborative Psychiatric Epidemiology Surveys 
After: NSAL was conducted as part of the Collaborative Psychiatric Epidemiology Surveys 
 Before: All participating adolescents Assent provided assent.
After: All participating adolescents provided assent. 
Before: NSAL-A sample was drawn from NSAL household
After: The NSAL-A sample was drawn from the NSAL household
Before: To measure income to needs ratio, participants parents/ guardians were asked about their family income was measured using self-reported data.
After: To measure income to needs ratio, participants parents/ guardians were asked about their family income using self-reported data.  -> 
Before: Arrows were drawn from independent variable and covariates to the mediator (% Whites) and outcome (perceived discrimination). 
After: Direct paths were assumed from independent variable and covariates to the mediator (% Whites) and outcome (perceived discrimination). 
Before: We used maximum likelihood to handle the missing data. 
After: We used full information maximum likelihood to handle the missing data. [Rev. I assume it was FIML: the sign is if the N in analysis does not drop at all or not much]
Before: Model 1 was ran in the pooled sample, with ethnicity and gender as covariates. Model 2 and Model 3 were estimated in each ethnic group, with gender as the covariate. Model 4 to Model 7 were conducted in each ethnicity by gender group. 
 After: Model 1 was tested on the pooled sample, with ethnicity and gender as covariates. Model 2 and Model 3 were estimated in each ethnic group, with gender as the covariate. Model 4 to Model 7 were conducted in each ethnicity by gender group. 
Before: In the pooled sample, higher family income to needs ratio was associated with higher % percentage of 
After: In the pooled sample, higher family income to needs ratio was associated with higher percentage of 

Before: Borrowing a national sample, 
After: Borrowing information from a national sample, 
Before: it is one of the first studies that suggest % Whites at school
After: it is one of the first studies that suggest the percent of Whites in schools
Before: Considerable research suggests that high SES increases exposure to discrimination for Blacks [].
After: Considerable research suggests that high SES increases exposure to discrimination for Blacks. [reference added]

Before: While most traditional theories such as 31 Fundamental Cause Theory (FCT) and Social Determinants of Health (SDH) theories [11–14] focus 32 on the health gain rather than the psychological costs that follows high SES.
After: Most traditional theories such as 31 Fundamental Cause Theory (FCT) and Social Determinants of Health (SDH) theories [11–14] focus 32 on the health gain rather than the psychological costs that follows high SES
Before: First, our study had a cross-sectional design, thus no causative inferences are allowed [52].
After: First, our study had a cross-sectional design, causative inferences depend on how well the models represent the causal processes under investigation [52].
For out tables, we mention the independent and dependent variables.

Figures will be fixed at the proof stage. They will appear in the paper larger, and will be expandable in the web page. Numbers are shown in the tables.

We fixed this sentence: "Conceptualization and theorization of The health disparities may benefit from instances that high SES becomes a risk factor."]
We have explained and reasoned why school composition is expected to have such a role.

Table 1 does not cut off on sides now.

Regarding the Income to needs ratio, we have cited the Census Buru which recommends this measurement.
We cited these citations:
Pearl, J. (1998). Graphs, causality, and structural equation models. Sociological Methods & Research, 27(2), 226. 
Pearl, J. (2011). The causal foundations of structural equation modeling. In R. H. Hoyle (Ed.), Handbook of structural equation modeling (pp. 68-91). New York, NY: The Guilford Press.]
 We revised our conclusion to make it more direct.
"for African American male youth, high SES may be associated with higher perceived discrimination because of attending schools with more Whites"

Round  2

Reviewer 1 Report

Please change the title to: Does School Racial Composition Explain Why High Income Black American Youth Perceive More Discrimination? A Gender Analysis